# Cohort profile: rationale and methods of UK Biobank repeat imaging study eye measures to study dementia

Paul J Foster [ORCID],[1] Denize Atan,[2] Anthony Khawaja [ORCID],[3] Andrew Lotery,[4] Tom MacGillivray,[5] Christopher G Owen,[6] Praveen J Patel,[7] Axel Petzold,[8,9] Alicja Rudnicka,[6] Zihan Sun [ORCID],[10] Simon Sheard [ORCID],[11] Naomi Allen,[11,12] On behalf of UK Biobank and UK Biobank Eye and Vision Consortium

For numbered affiliations see end of article.

**Correspondence to**
Dr Paul J Foster;
p.foster@ucl.ac.uk

## ABSTRACT

**Purpose** The retina provides biomarkers of neuronal and vascular health that offer promising insights into cognitive ageing, mild cognitive impairment and dementia. This article described the rationale and methodology of eye and vision assessments with the aim of supporting the study of dementia in the UK Biobank Repeat Imaging study.

**Participants** UK Biobank is a large-scale, multicentre, prospective cohort containing in-depth genetic, lifestyle, environmental and health information from half a million participants aged 40–69 enrolled in 2006–2010 across the UK. A subset (up to 60 000 participants) of the cohort will be invited to the UK Biobank Repeat Imaging Study to collect repeated brain, cardiac and abdominal MRI scans, whole-body dual-energy X-ray absorptiometry, carotid ultrasound, as well as retinal optical coherence tomography (OCT) and colour fundus photographs.

**Findings to date** UK Biobank has helped make significant advances in understanding risk factors for many common diseases, including for dementia and cognitive decline. Ophthalmic genetic and epidemiology studies have also benefited from the unparalleled combination of very large numbers of participants, deep phenotyping and longitudinal follow-up of the cohort, with comprehensive health data linkage to disease outcomes. In addition, we have used UK Biobank data to describe the relationship between retinal structures, cognitive function and brain MRI-derived phenotypes.

**Future plans** The collection of eye-related data (eg, OCT), as part of the UK Biobank Repeat Imaging study, will take place in 2022–2028. The depth and breadth and longitudinal nature of this dataset, coupled with its open-access policy, will create a major new resource for dementia diagnostic discovery and to better understand its association with comorbid diseases. In addition, the broad and diverse data available in this study will support research into ophthalmic diseases and various other health outcomes beyond dementia.

## INTRODUCTION

Dementia refers to a heterogeneous group of neurodegenerative disorders affecting 46.8 million people globally.[1–3] Alzheimer's disease (AD) is the most common, affecting 60%–80% of people with dementia.[4,5] Usually,

## STRENGTHS AND LIMITATIONS OF THIS STUDY

⇒ World's largest prospective, longitudinal multimodal imaging cohort with unprecedented power for analysis of determinants of a wide range of health outcomes.

⇒ Exceptional added value from the size, depth and quality of the cross-sectional and longitudinal MRI data on the eye, brain, body and imaging of heart, carotids, together with linkage to electronic health records, through which overt dementia and Alzheimer's disease can be identified.

⇒ Optimal timing to study cognitive impairment (age distribution: ~80% ≥60 years and ~65% ≥65 years).

⇒ Consistency of measurements between imaging devices over time, particularly with use of different optical coherence tomography devices.

⇒ Healthier participants compared with the general population.

a long prodromal period of up to 20 years of progressive cerebral atrophy is detectable on MRI scans and using body fluid biomarkers for neurodegeneration before AD is diagnosed.[2] These observations lead to a biological, rather than a clinical definition of AD.[6] To date, the majority of candidate drugs for slowing cognitive decline in AD or other dementias have failed in clinical trials,[7] probably because they are used too late in the natural history when irreversible, advanced degeneration has already set in.[8,9] Global roll-out of screening and disease progression monitoring strategies for AD based on MRI scans is precluded by their high cost and frequently limited availability. Body fluid biomarkers might provide ways of stratifying or diagnosing dementias but will remain complimentary to structural imaging biomarkers because of their lack of diagnostic specificity and are not recommended as a screening test.[10]

The eye provides insights into the risk or presence of some major systemic diseases,

including hypertension and diabetes, as well as changes associated with cognitive ageing and neurodegeneration.[11–13] As an alternative to MRI or plasma biomarkers, optical coherence tomography (OCT) offers a rapid, low-cost and non-invasive method for obtaining high-resolution (3–5 μm) images of the retina at the back of the eye—the only part of the central nervous system (CNS) that can be visualised directly. The laminated structure of the retina enables the direct monitoring of neurodegeneration at a near cellular level in vivo at a resolution that is more detailed than for any other non-invasive, in vivo imaging modalities. There is strong evidence that quantitative OCT measurements are associated with concurrent cognitive impairment and future cognitive decline and dementia.[14 15] In addition, OCT methods may directly monitor related vascular pathology: amyloid microangiopathy affects retinal and choroidal vasculature, as well as that in the cerebrum with AD.[16] Thus, retinal OCT scans offer the means to identify individuals at high risk of developing AD, providing them with opportunities to change their lifestyles or enter drug trials to delay or avert the onset of dementia. OCT scans are also a sensitive way to monitor patients for neurotoxic side effects of novel drug treatments. The ability to directly measure specific neuronal layers and microvascular characteristics in detail may provide a surrogate outcome marker for the CNS more generally and potentially enhance the power to detect disease much earlier than methods based on clinical history and genetic factors.

Several genetic (eg, *APOE*), comorbid (eg, diabetes, hypertension, depression, obesity) and lifestyle factors (eg, low educational attainment and smoking) have been associated with increased AD risk.[17–19] However, observational epidemiological studies cannot distinguish cause from effect and are vulnerable to bias from reverse causation and confounders. The analyses by Norton *et al* and Larsson *et al* illustrate how different disease risk factors can coexist and are often correlated, but they do not independently increase the risk of AD.[19 20] It also highlights the importance of identifying causal risk factors for dementia in designing upstream public health policies and social policies to reduce disease risk, clinical trials for new AD drugs and basic science research to understand the underlying mechanisms of dementia development. The large size of the cohort and the associated healthcare, imaging and genetic data make UK Biobank uniquely valuable for disambiguation of associations from causal comorbidities both for patient stratification and for elucidation of underlying mechanisms.

The research gaps, as mentioned earlier, motivated us to develop a major new resource for dementia diagnostic discovery and to better understand the association with comorbid diseases by adding rapid, low-cost OCT to the anticipated UK Biobank Repeat Imaging study, alongside ancillary testing of autorefraction/keratometry and fragmented letter test.[21] The objectives of this article are to describe (1) the process of test selection; (2) the methodology for eye and vision measures in the UK Biobank Repeat Imaging study and (3) the baseline characteristics of the study population in this study.

## COHORT DESCRIPTION
### UK Biobank
UK Biobank is a large-scale biomedical database and research resource containing in-depth genetic and health information from over 500 000 participants aged 40–69 enrolled across the UK between 2006 and 2010. Detailed study protocols are available on the UK Biobank website (https://www.ukbiobank.ac.uk/). It has become the pre-eminent biomedical research platform for studying the aetiology of common diseases of later life. During the baseline assessment, extensive sociodemographic, lifestyle and health-related information was collected through a touch screen questionnaire and oral interview, and a wide range of physical measurements was performed.[22 23] Participants also provided biological samples for genotyping, haematological, biochemistry, metabolomics and proteomics assays for the full cohort.[24]

UK Biobank is compliant with the previous Data Protection Act and the more recent General Data Protection Regulation (GDPR) implemented in 2018. For the GDPR, participants were contacted by email or post to explain how UK Biobank meets the requirements of the new regulations (https://www.ukbiobank.ac.uk/gdpr/).

At the baseline visit, ophthalmic assessments were performed on a subset of participants between 2009 and 2010 at 6 of 22 UK Biobank assessment centres, including visual acuity, autorefraction, keratometry, intraocular pressure, corneal biomechanics and retinal imaging comprising disc/macular digital colour photographs and a three-dimensional (3D) macular OCT. Over 110 000 participants have completed the visual acuity, refractive error and intraocular pressure measurements; and ~67 000 participants underwent retinal imaging. Detailed information on the baseline eye and vision measures has been published elsewhere.[22]

### The repeat imaging substudy
In 2014, UK Biobank launched the world's largest multimodal imaging study, intending to include baseline MRI of the brain, heart and abdomen, whole-body (dual-energy X-ray absorptiometry (DEXA) and carotid Doppler ultrasound on up to 100 000 participants. Detailed methods of the UK Biobank imaging enhancement were published elsewhere.[25] Although imaging 100 000 participants is a unique and powerful enhancement to the UKB resource, many valuable insights could only be gained from observing the change in imaging phenotypes over time. Recognising the importance of serial measurements, up to 60 000 of those in the imaging enhancement study will be invited to undergo repeat multimodal imaging between 2022 and 2028. As part of the repeat imaging study, data collection of the eye measures (eg, OCT) is anticipated to take place from 2022 to 2028. The specific study design is as follows:

- All UK Biobank participants who have previously attended a baseline brain and body imaging visit will be invited to attend a repeat imaging visit (the invitation will specify the same imaging centre as their baseline imaging visit to minimise measurement error caused by differences between scanners at different centres).
- Appointment slots are planned in groups of 3 to minimise equipment downtime and to maximise participant throughput and data quality.
- On arrival, those who accept will be asked to consent to the study, and each participant will then undergo a pre-screening safety assessment.
- The three participants would then progress to the imaging modalities as follows:
  - Participant #1≥brain MRI.
  - Participant #2≥abdomen and heart MRI.
  - Participant #3≥DEXA, ultrasound and OCT.
- Each imaging modality 'group' takes approximately 40 min, after which the participants will move to the next modality, then switch again after 40 min so that each member of the group of three has visited all three imaging measurement stations over a 2-hour period.
- The participants then all progress to the non-imaging parts of the visit where they will complete questionnaires, have physical measures and give biological samples, which mirrors much of the initial (2006–2010) baseline visit.
- As one group of three participants exits the imaging part of the visit, the next group of three are ready to enter, thus ensuring that the imaging part of the visit is fully used.
- This process will repeat for five groups of 3 people (15 total) on 7 days per week at each of UK Biobank's 4 dedicated imaging centres.

## Study location

This multisite study will be run from four dedicated UK Biobank Imaging Centres across the UK (Newcastle upon Tyne, Stockport, Reading and Bristol). These four centres help ensure most participants are within a reasonable distance to attend a scanning visit. As far as is reasonably practical, maintaining the same instruments and software/firmware across the sites and all phases of the UK Biobank project will ensure consistency and comparability of results from the start of the baseline imaging project to the end of the repeat imaging programme. UK Biobank built the following strategies to reduce variability across the different sites:

- The sites are each populated with the same equipment (same manufacturer, same model, same software/firmware, etc) configured with the same protocols and the same settings.
- MR scanner settings/performance across all four sites is monitored by UK Biobank's full-time in-house MR physicist with continuous quality assurance processes to identify and resolve quality issues that may arise.

- All staff are trained to standard operating procedures, and (in the case of the imaging element) compliance/consistency is overseen by an in-house senior radiographer and an in-house MR physicist. The non-imaging aspects are overseen by UK Biobank's dedicated 'Training and Monitoring' team.
- Systems are already in place to ensure appropriate levels of training for all operational staff, monitored via Clinic Training Assessments/Training Matrices. These will be extended to cover the OCT measures: appropriate training will be provided, training assessments/matrices extended to cover these measures, and performance monitored.
- Imaging data are routinely made available to members of the project's expert working group, which is made up of experts in each of the imaging modalities; this group monitors the project progress, periodically provides training interventions and critically, periodically/routinely provides an independent view of performance and data output. A similar approach will be taken regarding the OCT measures with data made available to the Eye Consortium members listed in this application for quality control purposes.

## Recruitment

The UK Biobank cohort includes a committed and engaged group of participants who are regularly invited for follow-up activities: the typical response rate to online surveys is >50%, and there have been very few withdrawals from the study since recruitment (<0.2%). Regular communications with the cohort (via newsletters, participant meetings, study update meetings and the participant section of the UK Biobank website: www.ukbiobank.ac.uk/explore-your-participation) help to maintain enthusiasm for and engagement with the study. Direct telephone communication with individual participants regarding new substudies or general participation questions via a dedicated 'Participant Contact Centre' provides personalised information and reassurance.

This study will use the same invitation protocol as the UK Biobank imaging enhancement study (2014–2023).[25] The planned protocol for this repeat imaging study involves:

- Email/postal explanation of the study and invitation to book an appointment.
- A telephone call to book an appointment and perform safety prescreening via participant resource centre.
- Assessment at the nearest of four imaging centres across the UK (Stockport, Newcastle, Reading and Bristol) to minimise travel time and maximise participant attendance.

At the start of the pandemic (lockdown in the UK in March 2020), 50 000 of the target 100 000 participants had been imaged. Participant questionnaires on completion of baseline imaging visit indicate >90% would be happy to undertake a repeat imaging visit. Pilot studies involving a few thousand participants have demonstrated

~60% acceptance rates, providing confidence that ~60 000 could be recruited for this repeat imaging study.

## Examination procedures

The whole-body imaging modalities have been extensively detailed elsewhere[25]; this article only describes the scope of eye and vision measurements in the Repeat Imaging study. The Topcon Triton OCT platform is being used to obtain OCT images in this study. The Triton platform uses ultra-highspeed swept-source (SS) OCT technology with a central wavelength of 1050 µm that penetrates deeper than the retina, allowing visualisation of the choroid and the vasculature therein.[26] The platform also takes colour retinal fundus photographs immediately after the OCT scans, allowing measurement of the optic disc and retinal vessel metrics (including retinal vessel calibre and tortuosity). The Topcon Triton supports wide-angle 12 mm×9 mm scans that include the optic disc and macula in a single scan.

Widefield SS-OCT enables quantitative measurements of several candidate biomarkers, including but not limited to total macular retinal thickness, macular inner retinal sublayer thicknesses, peripapillary retinal nerve fibre layer thickness, choroidal vascularity index, retinal arteriolar and retinal venular calibres, retinal vascular fractal dimension, retinal vascular tortuosity. Details of the candidate biomarkers are summarised in table 1.

## OCT image processing

Total retinal thickness and segmented values for retinal sublayer thicknesses for macula and optic nerve scans are generated by the current generation OCT devices, using US Food & Drug Administration (FDA) approved algorithms, during the examination. In contrast to the processing of baseline UK Biobank macular OCT scans,[27] they do not generally require the development of new processing pipelines (apart from measures of the choroidal vascular layer, which are now possible thanks to greater depth of imaging than was previously possible with older OCT technology). Fundamental to both the challenge and the opportunity that would be provided by UK Biobank OCT imaging is that modern retinal imaging software can measure changes that would be imperceptible to, or missed by, a human grader. In operations featuring large-scale data collection, a small proportion of the imaging is likely to be insufficient quality for automated analysis. Problems may also arise with image acquisition, for instance, due to some study participants presenting with ocular pathology. Thus, the first step in an analysis pipeline is to assess the image quality and discard images that cannot be adequately measured. Subsequent analysis of vasculometrics from retinal photographs would include automated vessel segmentation, followed by classification of arterioles and venules (see comments on QUARTZ and VAMPIRE vasculometry systems in the next 2 paragraphs). For OCT, algorithms delineate the borders of the internal limiting membrane (ILM) and the RNFL to give the measurement of RNFL thickness, a

**Table 1** Description of the candidate biomarkers

| Biomarkers | View | Description |
| --- | --- | --- |
| Total macular retinal thickness | Cross-section | Distance between the inner boundary of ILM to the lower boundary to RPE |
| Macular inner retinal sublayer thicknesses | | |
| RNFL thickness | Cross-section | Distance between ILM to the outer boundary of RNFL |
| GC-IPL thickness | Cross-section | Distance between the inner boundary of GCL to the outer boundary of IPL |
| GCC thickness | Cross-section | GC-IPL+RNFL |
| Peripapillary RNFL thickness | Cross-section | Distance between ILM to the outer boundary of RNFL |
| Choroidal vascularity index | Cross-section/en face | Ratio of vascular luminal area to the total choroidal area |
| Retinal arteriolar calibres | En face | Evaluates generalised arteriolar narrowing |
| Retinal venular calibres | En face | Evaluates generalised arteriolar narrowing |
| Retinal vascular fractal dimension | En face | Measure the vascular pattern complexity |
| Retinal vascular tortuosity | En face | Characterised by an abnormal curvature of the vessels, evincing a non-smooth appearance, presenting turns and twists throughout their course. |

GCC, ganglion cell complex; GCIPL, ganglion cell-inner plexiform layer; GCL, ganglion cell layer; ILM, inner limiting membrane; IPL, inner plexiform layer; m, macular; RNFL, retinal nerve fibre layer; RPE, retinal pigment epithelium.

biomarker of axonal loss affecting retinal ganglion cells and the optic nerves. The thickness of the RNFL is evaluated using the standard TSNIT (temporal, superior, nasal, inferior, temporal) mapping that subdivides the measurements and colour codes statistical significance compared with a database of normal healthy values. Further delineation of boundaries enables quantitative mapping of the ganglion cell layer and inner plexiform layer thicknesses, a marker of neuronal somatic loss.[13 28]

Although the processing of quantitative retinal vasculometric data is not routinely used in clinical settings, we have developed and validated a fully automated AI-enabled retinal image analysis system (QUARTZ) for extracting vessel maps and quantifying retinal vasculometry (including vessel size and tortuosity), which we will use to create the image processing pipeline. The system overcomes many of the difficulties of earlier vasculometry approaches, particularly by being fully automated.[29 30]

QUARTZ has been demonstrated to be highly robust, capable of processing large datasets with automated image quality assessments, resulting in accurate, reliable and high levels of vessel segmentation. To date, QUARTZ has measured approximately 4 million vessel segments from over 190 000 images from 95 000 participants of two very large population-based cohorts (UK Biobank and EPIC-Norfolk). This system has been developed specifically for use on TOPCON macular centred images.

In brief, the QUARTZ system distinguishes between right and left eyes, venules and arterioles (with 87% accuracy using AI-enabled deep learning), identifies vessel segments and centreline coordinates and outputs measures of vessel width and tortuosity (based on the mean change in chord length between successive divisions of the vessel).[31 32] The system obtains 10–20 thousands of measurements of width and tortuosity from the whole retinal image (dependent on image quality), not just selected vessels lying within concentric areas centred on the disc. Measures are summarised using mean width and tortuosity, weighted by segment length, for arterioles and venules separately for each image. QUARTZ measures in UK Biobank have previously shown that venular width and tortuosity are associated with markers of adiposity[33] and that both arteriolar and venular width and arteriolar tortuosity show strong inverse associations with blood pressure (systolic and diastolic) and arterial stiffness index.[34] More importantly, prognostic models using QUARTZ vasculometry measures perform very well at predicting circulatory mortality and at least as well as established risk scores in the prediction of stroke and myocardial infarction, remarkably without the need for either a blood test or blood pressure measurement.[35] Given the identification of vessel maps, these could be inputted into other systems (ie, the Vascular Assessment and Measurement Platform for Images of the REtina, VAMPIRE system) with additional vasculometry summaries, such as fractal analyses to quantify the complexity of the arteriolar and venular components of the retinal vascular network. Marrying the automated functionality of QUARTZ with VAMPIRE will afford a more in-depth characterisation of the vessel complex on an unprecedented scale.

The VAMPIRE system is an international collaborative project designed to quantify retinal vascular morphometry with large collections of fundus photographs. The system provides automatic detection of retinal landmarks and quantifies some key parameters used frequently in investigative studies—vessel width, vessel branching coefficients, tortuosity and fractal analyses. Detailed definitions have been reported elsewhere.[36 37] In general, it computes 149 measurements per image, including basic statistics. Thirty-nine are width related: central retinal arteriolar equivalent, central retinal venular equivalent, retinal arteriovenous ratio, basic statistics (mean, median, SD, maximum, minimum), width gradients along vessels, average ratio length diameter at branching points, by arteries and veins; 104 are tortuosity measurements,

computed by different algorithms and with the statistics listed above; 6 are fractal dimension coefficients. All measures are calculated by vessel type (arteriole or venule) and region (zone, whole image, quadrants). VAMPIRE is a validated software application and has been extensively used in several international studies.[36 38 39]

## Patient and public involvement
UK Biobank maintains a website to keep participants and researchers up to date on the study (http://www.ukbiobank.ac.uk/news/). Eye and vision-related publications resulting from UK Biobank are maintained at (https://www.ukbiobankeyeconsortium.org.uk/publications). UK Biobank also holds regular events to inform the participants about the imaging study and the latest research. In addition, UK Biobank has a Twitter feed (@uk_biobank). The study was set up by the Medical Research Council, Department of Health and Wellcome Trust with input from major patient representative organisations. An annual scientific meeting is recorded and available to the public as a webcast.

## Statistical analysis plan
Baseline ocular characteristics will be summarised as mean (SD) for continuous variables and number (%) for categorical variables.

Primary aims would be to examine:
1. Cross-sectional associations between retinal biomarkers, measures of cognitive performance and brain-volume from MRI imaging.
2. The comparative performance of retinal biomarkers for risk stratification, to identify those with cognitive impairment.
3. The comparative performance of retinal biomarkers to detect those with longitudinal decline in cognitive performance.

Previous work within UK Biobank examining RNFL measures in relation to mild cognitive impairment showed that those in the lowest quintile of RNFL thickness were 11% (95% CI 2% to 21%) more likely to fail on at least one of four cognitive tests.[40] This shows that RNFL measures have the potential to identify those at higher risk of cognitive impairment. After vigorous image quality control, the proposed imaging of a further 60 000 participants will provide 45 000–55 000 participants with good-quality retinal images for quantification of individual components of the RNFL and potential to extract detailed retinal vasculometric measures. This large sample size, will have 99% power (alpha=0.001) to detect at least 0.03 SD change in the cognitive score[41] or brain measures[42] (based on F-tests of linear regression coefficients from cross-sectional analyses) per 1 SD increase in any retinal biomarker (RNFL or retinal vasculometric measure). Cross-sectional analyses using multiple linear regression will quantify the dose–response relationship between cognitive score with considerable power to evaluate in the region of 30 candidate predictors (retinal

**Table 2** Demographics, cognitive scores, *APOE* genotype, self-reported comorbidities, medication use and availability of eye imaging factors for UK Biobank participants attending the baseline imaging assessment to date (N=48 998)

| Characteristics | n (%) or mean (SD) |
|---|---|
| Age (years) | 55.2 (7.6) |
| Sex | |
| Female | 25 290 (52%) |
| Male | 23 708 (48%) |
| Cognitive scores at baseline assessment | |
| Numeric memory: maximum digits remembered correctly (n=4911) | 6.97 (1.25) |
| Fluid intelligence score (n=16 427) | 6.68 (2.04) |
| Prospective memory test (n=16 544) | 1.10 (0.36) |
| Snap game: mean time to correctly identify matches (ms) (n=48 858) | 539.2 (101.3) |
| Pairs matching: number of incorrect matches in round (n=24 988) | 0.66 (1.24) |
| APOE genotype | |
| $\varepsilon3\varepsilon3$ | 28 297 (59%) |
| $\varepsilon3\varepsilon4$ | 11 063 (23%) |
| $\varepsilon2\varepsilon3$ | 5892 (12%) |
| $\varepsilon2\varepsilon4$ | 1128 (2%) |
| $\varepsilon4\varepsilon4$ | 1065 (2%) |
| $\varepsilon2\varepsilon2$ | 277 (1%) |
| Comorbidities | |
| Hypertension | 13 666 (28%) |
| Diabetes | 2008 (4%) |
| Ischaemic heart disease | 2649 (5%) |
| Stroke | 719 (1%) |
| Chronic obstructive pulmonary disease | 1016 (2%) |
| Asthma | 6689 (14%) |
| Obesity (BMI>30 kg/m$^2$) | 8918 (18%) |
| Parkinson's disease | 136 (<1%) |
| Alzheimer's disease | 34 (<1%) |
| Multiple sclerosis | 202 (<1%) |
| Medication use | |
| Anti-hypertensive | 9769 (20%) |
| Statin | 8750 (18%) |
| Eye imaging available | 13 732 (28%) |

Mean (SD) is presented for continuous variables and count (%) for categorical variables. All variables are presented for the full sample except for APOE genotype (1276 missing) and for cognitive scores (numbers of participants for each test at any phase of UK Biobank examinations are presented in the table). APOE, apolipoprotein E; BMI, body mass index.

biomarkers, age, sex, geographical location, height, refraction, intraocular pressure, smoking status, socio-economic positions and established cardiovascular risk markers).[43] This will allow the independent contribution of retinal biomarkers as a predictor of cognitive performance to be realised with considerable precision,[44] across a spectrum of cognitive scores.[43]

Given that UK Biobank has longitudinal data on cognitive change (with repeated measures available from online questionnaires and performed in-person at the imaging assessments), the study would be uniquely placed to assess the determinants of cognitive decline in middle-later life. For prospective evaluation the rates of dementia would also be pivotal in relation to prior cognitive performance. In UK Biobank, the annual incidence of dementia among those aged ≥60 years old is approximately 2.5 per 1000 person years.[45] Therefore, within 2 years of retinal image capture there would be approximately 250 cases of dementia per 45 000–55 000 participants. The longitudinal nature of the data will allow models to be developed for incident cognitive outcomes/neurodegenerative events using multivariable Cox proportional hazards models with relevant eye measures (ie, OCT, retinal vasculometry derived measures) as continuous predictors both with and without inclusion of other parameters, including age at cognitive decline/neurodegenerative onset, sex, ethnicity (although the cohort is largely of white European ancestry), smoking status (current, former and never), alcohol consumption, body mass index, blood pressure, blood biochemistry measures, social deprivation (by postal code), physical activity/sedentary behaviour and relevant family history where available.

### Existing data
Once recruitment was fully under way, additional measures were incorporated into the baseline assessment, including hearing and arterial stiffness tests, a cardiorespiratory fitness test and various eye and vision measures, including visual acuity on a computerised system designed to observe logarithm of the minimum angle of resolution principles, and following the British Standard (BS-1968),[46] autorefraction and keratometry, intraocular pressure and corneal biomechanics, and retinal imaging comprising disc/macular digital colour photographs and a 3D macular OCT.[22] After the baseline visit, subsets of participants have supported additional data collection through various enhancements to the study. These have included: a complete repeat of the baseline assessment, collection of physical activity data over 7 days by wearing accelerometers, and regular online questionnaires covering various topics such as diet, cognitive function, occupational history, mental well-being, gastrointestinal health and pain. All participants provided consent for their health to be followed-up through linkage to health-related records, which currently includes death, cancer and hospital inpatient records for the entire cohort. Although UK Biobank is not representative of the entire UK population, the large sample size and variation across all levels of measures nonetheless enable a valid assessment of many exposure–outcome relationships to be made. All publications using UK Biobank data are available on the

**Table 3** Cognitive function measures currently available in UK Biobank at different time points

| (Variable ID) | Study phase (n) | | | | |
| --- | --- | --- | --- | --- | --- |
| | Baseline (n) (2006–2010) | Repeat assessment (2012–2013) | Online (2015) | Imaging study (2014–now) | Repeat imaging (2019–2020) |
| Fluid IQ (100027) | 165 500 | 20 100 | 123 500 | 39 600 | 800 |
| Pairs matching (100030) | 497 900 | 20 300 | 118 500 | 40 400 | 800 |
| Prospective memory (100031) | 171 600 | 20 300 | 0 | 40 400 | 800 |
| Reaction time (100032) | 496 700 | 20 300 | 0 | 40 200 | 800 |
| Numeric memory (100029) | 51 800 | 0 | 111 000 | 28,7000 | 800 |
| Matrix (501) | 0 | 0 | 0 | 27 600 | 800 |
| Symbol digit substitution (502) | 0 | 0 | 118 500 | 27 600 | 800 |
| Tower test (503) | 0 | 0 | 0 | 27 300 | 800 |
| Picture vocabulary (504) | 0 | 0 | 0 | 27 500 | 800 |
| Trail making (505) | 0 | 0 | 120 500 | 27 900 | 800 |
| Paired associate learning (506) | 0 | 0 | 0 | 27 900 | 800 |

website (https://www.ukbiobank.ac.uk/enable-your-research/publications). Eye and vision-related publications resulting from UK Biobank is maintained at (https://www.ukbiobankeyeconsortium.org.uk/publications).

In brief, based on data from UK Biobank participants attending the baseline imaging assessment to date (N=48 998), the mean (SD) age was 55.2 (7.6) years; 52% (N=25 290) of them were female. A subset of 13 732 (28%) participants had undergone retinal imaging. As there is a policy for the UK Biobank Repeat Imaging Study to oversample participants with baseline retinal imaging, the estimated numbers of participants with overlapping retinal imaging and whole-body imaging data in the repeat imaging visit will be more than 16 800. Detailed cognitive scores, APOE genotypes, self-reported comorbidities and medication use are provided in table 2. In addition to imaging, UK Biobank has implemented a wide range of cognitive function tests since baseline that are relevant to assessing various aspects of cognitive decline and dementia and will be conducted at the repeat imaging and proposed OCT visit (table 3).

## FINDINGS TO DATE

UK Biobank has helped make significant advances in the understanding of risk factors for diseases including cardiovascular diseases, cancer, diabetes, stroke, multiple sclerosis, optic neuritis and dementia.[23 47–57] Ophthalmic genetics and epidemiology have benefited from the unparalleled combination of very large numbers of participants, very extensive and detailed phenotyping and longitudinal follow-up.[30 58–62]

In addition, we have used UK Biobank data to describe the relationship between retinal structures and both cognitive function and brain MR image-derived phenotypes.[40 42] For example, previous work examining RNFL measures in relation to mild cognitive impairment, showed that those in the lowest quintile of RNFL thickness were 11% (95% CI 2.0% to 2.1%) more likely to fail on at least one of four cognitive tests.[40] This indicates that RNFL thickness measurements have the potential to identify those at higher risk of cognitive impairment. Chua *et al*[42] reported that markers of retinal neurodegeneration are associated with smaller brain volumes—macular ganglion cell-inner plexiform layer thickness, ganglion cell complex (GCC) thickness and total macular thickness were significantly associated with smaller total brain (p<0.001), grey matter and white matter volume (p<0.01), and grey matter volume in the occipital pole (p<0.05); thinner macular GCC and total macular thicknesses were associated with smaller hippocampal volume (p<0.02).

In the context of these results, and the findings of other studies (eg, The Rotterdam Study),[63–65] we proposed supplementing the testing menu in the UK Biobank Whole Body Repeat Imaging Study with measures that support the discovery and quantification of eye and vision variables that are associated with cognitive ageing and decline, and overt dementia.

## COLLABORATION

UK Biobank aims to provide open access data for healthcare-related research. The data are available to all bona fide researchers from the academic, charity, public and commercial sectors in the UK and internationally, without preferential or exclusive access for any user.[66] All interested researchers may apply to access the data via an online application. Strict guidelines are in place to help ensure anonymity and confidentiality of participants' data and samples.[67] We have formed the UK Biobank Eye and Vision Consortium, an 80 person strong group of researchers with interest and expertise in ophthalmic epidemiology, visual system neurology and the epidemiology of related diseases such as diabetes and cardiovascular disease (https://www.ukbiobankeyeconsortium.org.uk/).

**Author affiliations**
[1]Moorfields Eye Hospital NHS Foundation Trust, NIHR Moorfields Biomedical Research Centre, London, UK
[2]Medical School, University of Bristol, Bristol, UK
[3]NIHR Biomedical Research Centre at Moorfields Eye Hospital NHS Foundation Trust & UCL Institute of Ophthalmology, London, UK
[4]Faculty of Medicine, University of Southampton, Southampton, UK
[5]Clinical Research Imaging Centre, Queens Medical Research Institution, Edinburgh, UK
[6]Population Health Research Institute, St Georges Medical School, University of London, London, UK
[7]NIHR Biomedical Research Centre, Moorfields Eye Hospital NHS Foundation Trust, London, UK
[8]Department of Molecular Neurosciences, Moorfields Eye Hospital and The National Hospital for Neurology and Neurosurgery, Queen Square Institute of Neurology, UCL, London, UK
[9]Departments of Neurology, Ophthalmology and Expertise Center for Neuro-ophthalmology, Amsterdam UMC, Amsterdam, The Netherlands
[10]Institute of Ophthalmology, University College London, London, UK
[11]UK Biobank, Stockport, UK
[12]Nuffield Department of Population Health, University of Oxford, Oxford, UK

**Acknowledgements** In addition to the listed authors, Prof Rory Collins, Prof Paul Matthews and Dr Mark Effingham participated in scientific discussions which moulded the project that we have outlined here. A funding proposal was developed following discussions with members of the Alzheimer's Drug Discovery Foundation. Meanwhile, we would like to thank all the participants of UK Biobank for their vital contribution to the resource.

**Collaborators** UK Biobank Eye and Vision Consortium: Professor Naomi Allen—University of Oxford, Professor Tariq Aslam—The University of Manchester, Dr Denize Atan—University of Bristol, Dr Konstantinos Balaskas—Moorfields Eye Hospital, Professor Sarah Barman - Kingston University, Professor Jenny Barrett—University of Leeds, Professor Paul Bishop—The University of Manchester, Professor Graeme Black—The University of Manchester, Dr Tasanee Braithwaite—St Thomas' Hospital, Dr Roxana Carare—University of Southampton, Professor Usha Chakravarthy—Queen's University Belfast, Dr Michelle Chan—Moorfields Eye Hospital, Dr Sharon Chua—UCL Institute of Ophthalmology, Dr Alexander Day - Moorfields Eye Hospital, Dr Parul Desai—Moorfields Eye Hospital, Professor Bal Dhillon—University of Edinburgh, Professor Andrew Dick—University of Bristol, Dr Alexander Doney—University of Dundee, Dr Cathy Egan—Moorfields Eye Hospital, Professor Sarah Ennis - University of Southampton, Professor Paul Foster—UCL Institute of Ophthalmology, Dr Marcus Fruttiger - UCL Institute of Ophthalmology, Dr John Gallacher—University of Oxford, Professor David Garway-Heath—UCL Institute of Ophthalmology, Dr Jane Gibson—University of Southampton, Professor Jeremy Guggenheim—Cardiff University, Professor Chris Hammond—King's College London, Professor Alison Hardcastle—UCL Institute of Ophthalmology, Professor Simon Harding—University of Liverpool, Dr Ruth Hogg—Queen's University Belfast, Dr Pirro Hysi—King's College London, Professor Pearse Keane—UCL Institute of Ophthalmology, Professor Sir Peng Tee Khaw—UCL Institute of Ophthalmology, Dr Anthony Khawaja—Moorfields Eye Hospital, Gerassimos Lascaratos—Moorfields Eye Hospital, Dr Thomas Littlejohns—University of Oxford, Professor Andrew Lotery - University of Southampton, Dr Robert Luben—UCL Institute of Ophthalmology, Professor Phil Luthert—UCL Institute of Ophthalmology, Dr Tom Macgillivray—University of Edinburgh, Dr Sarah Mackie—University of Leeds, Dr Savita Madhusudhan—Royal Liverpool University Hospital, Dr Bernadette Mcguinness - Queen's University Belfast, Dr Gareth Mckay—Queen's University Belfast, Dr Martin Mckibbin—Leeds Teaching Hospitals NHS Trust, Professor Tony Moore—UCL Institute of Ophthalmology, Professor James Morgan—Cardiff University, Dr Eoin O'Sullivan—King's College Hospital, Professor Richard Oram—University of Exeter, Professor Chris Owen—St George's, University of London, Dr Praveen Patel—Moorfields Eye Hospital, Dr Euan Paterson - Queen's University Belfast, Dr Tunde PETO—Queen's University Belfast, Dr Axel PETZOLD—UCL Institute of Neurology, Dr Nikolas PONTIKOS—UCL Institute of Ophthalmology, Professor Jugnoo RAHI—UCL Institute of Child Health, Dr Alicja RUDNICKA—St George's, University of London, Professor Naveed Sattar—University of Glasgow, Dr Jay SELF—University of Southampton, Dr Panagiotis SERGOUNIOTIS—The University of Manchester, Professor Sobha SIVAPRASAD—Moorfields Eye Hospital, Professor David STEE—Newcastle University, Ms Irene STRATTON—Gloucestershire Hospitals NHS Foundation Trust, Dr Nicholas STROUTHIDIS—Moorfields Eye Hospital, Professor Cathie SUDLOW—University of Edinburgh, Dr Zihan SUN—UCL Institute of Ophthalmology, Dr Robyn TAPP—St George's, University of London, Dr Dhanes THOMAS—Moorfields Eye Hospital, Professor Emanuele TRUCCO—University of Dundee, Dr Adnan TUFAIL—Moorfields Eye Hospital, Dr Ananth VISWANATHAN—Moorfields Eye Hospital, Dr Veronique VITART—University of Edinburgh, Dr Mike WEEDON—University of Exeter, Dr Katie WILLIAMS—King's College London, Professor Cathy WILLIAMS—University of Bristol, Professor Jayne WOODSIDE—Queen's University Belfast, Dr Max YATES—University of East Anglia, Dr Yalin ZHENG—University of Liverpool.

**Contributors** PJF, AK, PJP and ZS had full access to all the data in the study and took responsibility for the integrity and accuracy of the data analysis. Concept and design: PJF, DA, AK, AL, TM, CGO, PJP, AP and AR. Data acquisition, analysis or interpretation: UK Biobank obtained the data. AK performed data analysis. All authors interpreted data. Critical revision of the manuscript for important intellectual content: all authors. Obtained funding: NA, SS and UK Biobank. All authors approved the final manuscript. PJF is the guarantor and accepts full responsibility for the report and controlled the decision to publish.

**Funding** The study sponsor/funder was not involved in the design of the study; the collection, analysis and interpretation of data; writing the report; and did not impose any restrictions regarding the publication of the report. UK Biobank is funded by the Medical Research Council, Wellcome Trust, Department of Health, Scottish Government, the Welsh Assembly Government, British Heart Foundation, Cancer Research UK, NIHR and the Northwest Regional Development Agency. The UK Biobank Eye and Vision Consortium is supported by grants from Moorfields Eye Charity, the NIHR Biomedical Research Centre at Moorfields Eye Hospital NHS Foundation Trust and UCL Institute of Ophthalmology, the Alcon Research Institute and the International Glaucoma Association (UK). AK, AP, ZS, PJF and PJP receive salary support from the NIHR BRC at Moorfields Eye Hospital & UCL Institute of Ophthalmology. NA receives salary support from University of Oxford and UK Biobank. PJF receives support from the Desmond Foundation, London, UK. AK is supported by a UKRI Future Leaders Fellowship and an Alcon Research Institute Young Investigator Award. TM acknowledges support from NHS Lothian R&D and the Clinical Research Facility at the University of Edinburgh. The authors acknowledge a proportion of our financial support from the UK Department of Health through an award made by the National Institute for Health Research to Moorfields Eye Hospital NHS Foundation Trust and UCL Institute of Ophthalmology for a Biomedical Research Centre for Ophthalmology.

**Competing interests** PJF reports personal fees from Allergan, Carl Zeiss, Google/DeepMind and Santen, a grant from Alcon, outside the submitted work. PJP reports grants from Topcon Inc, outside the scope of the current report. AK reports personal fees from Abbvie, Aerie, Google Health, Novartis, Reichert, Santen and Thea, outside the submitted work. AP reports grant support for remyelination trials in multiple sclerosis to the Amsterdam University Medicam Centre, Department of Neurology, MS Centre (RESTORE trial) and UCL, London RECOVER trial; Fight for Sight (nimodipine in optic neuritis trial); royalties or licenses from Up-to-Date (Wolters Kluver) on a book chapter; speaker fees for the Heidelberg Academy; participation on Advisory Board SC Zeiss OCTA Angi-Network, SC Novartis OCTiMS study; equipment: OCTA from Zeiss (Plex Elite).

**Patient and public involvement** Patients and/or the public were involved in the design, or conduct, or reporting, or dissemination plans of this research. Refer to the Methods section for further details.

**Patient consent for publication** Not applicable.

**Ethics approval** This study involves human participants and UK Biobank received approval from the National Information Governance Board for Health and Social Care and the National Health Service Northwest Centre for Research Ethics Committee (Ref: 11/NW/0382).

**Provenance and peer review** Not commissioned; externally peer reviewed.

**Data availability statement** Data are available on reasonable request. Data are available globally to bona fide researchers and scientists from UK Biobank, and completion of registration and a data access request.

ORCID iDs
Paul J Foster http://orcid.org/0000-0002-4755-177X
Anthony Khawaja http://orcid.org/0000-0001-6802-8585
Zihan Sun http://orcid.org/0000-0002-5001-7113
Simon Sheard http://orcid.org/0000-0002-8925-8280

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
