## [Reviewer comments · BMJ Open]

ARTICLE DETAILS

TITLE (PROVISIONAL)	Cohort Profile: Rationale and Methods of UK Biobank Repeat Imaging Study Eye Measures to Study Dementia
AUTHORS	Foster, Paul; Atan, Denize; Khawaja, Anthony; Lotery, Andrew; MacGillivray, Tom; Owen, Christopher; Patel, Praveen; Petzold, Axel; Rudnicka, Alicja; Sun, Zihan; Sheard, Simon; Allen, Naomi

VERSION 1 – REVIEW

REVIEWER	Samantha Lee Lions Eye Institute, Centre for Ophthalmology and Visual Science, University of Western Australia
REVIEW RETURNED	28-Dec-2022

GENERAL COMMENTS	This is a generally well-written report and a very important follow-up study to conduct. The methodology and protocol are very well thought out and reported. I'm excited to see where this study leads to in terms of dementia and eye disease outcomes. I only have very minor comments. My main comment is that the study purpose deviated from my expectation based on the title. The title and the purpose in the abstract suggest that the aim of this study was to collect and examine the eye measures for the purpose of studying eye diseases. But the Findings to Date section in the abstract took an odd and unexpected turn to talk about dementia. The introduction also starts out with dementia (imagine a reader reading the main text immediately after the title, it can get confusing) and only in the second paragraph it is clear that the eye imaging is mainly used for the dementia research. However, OCT is not used in dementia diagnosis, as the authors mentioned in the second paragraph. I suggest modifying the title to reflect the focus on dementia and/or make some changes to the introduction to manage readers' expectations. While it is implied in the title, perhaps indicate in the abstract whether the subset of participants returning had a baseline imaging. Also, which years the baseline imaging was conducted and whether the age (40–69) refers to baseline age. Line 54 abstract: "... in 2022–2028." Last sentence of abstract and strengths of study: If spaces allows, I think the authors can afford to be a bit more optimistic. The repeat imaging studies are likely to benefit research in ophthalmic diseases and other health outcomes. Introduction "The eye provides insights into the risk or presence of
--

	all systemic diseases...” – I’m not sure this is accurate. It may be true for most major diseases but it’s hard to prove for “all” diseases (e.g. cancers?). “Using quantitative measurements from OCT scans, it may be possible to assess causal relationships between risk factors and retinal biomarkers related to dementia.” – It is not clear sure how OCT scans of the eye, even if of the retinal ganglion cells/nerve fibres or the retinal vasculature, can help assess these relationships. I’m assuming that the OCT measures will be used as surrogates of dementia? However, this will have the problem of factors that locally affect the ocular structures (e.g. IOP, refractive error) which are unlikely to impact the systemic outcome. It may be worth mentioning how the subset of the participants who underwent eye testing at baseline were selected (line 150). It is also unclear whether the 60,000 return participants have had a baseline OCT. It would be interesting to see how longitudinal change in retinal biomarkers is associated with cognitive measures and incidence of dementia.
--	--

REVIEWER	Helmut Kuechenhoff Ludwig-Maximilians-Universitat Munchen, Department of Statistics
REVIEW RETURNED	23-Feb-2023

GENERAL COMMENTS	The UK Biobank is an important and well designed study and extremely useful for many types of research. However, the part of statistical analysis should contain more concrete strategies of modelling those data. There are basically no strategies for longitudinal analysis of the data (p. 17) At least some basic ideas of longitudinal data analysis could be given in some detail. Furthermore power calculations on page 16 were made without mentioning the relating statistical test (t-test ?).
--

VERSION 1 – AUTHOR RESPONSE

Reviewer #1 Dr. Samantha Lee, Lions Eye Institute Comments to the Author:		
Suggestion, Question, or Comment from Reviewer #1	Author’s Response	Change in the Manuscript
This is a generally well-written report and a very important follow-up study to conduct. The methodology and protocol are very well thought out and reported. I’m excited to see where this study leads to in terms of dementia and eye disease outcomes. I only have very minor comments.	We thank the reviewer for the positive feedback on the report. We agree that the title might be misleading. We will revise the title and the purpose of the abstract and highlight that our focus is on dementia research.	(Lines 1-2) Title Original: Cohort Profile: Rationale and Methods of UK Biobank Repeat Imaging Study Eye Measures Revised: Cohort Profile: Rationale and Methods of UK Biobank Repeat Imaging Study Eye Measures to Study Dementia (Lines 36-41) Purpose section of the abstract

My main comment is that the study purpose deviated from my expectation based on the title. The title and the purpose in the abstract suggest that the aim of this study was to collect and examine the eye measures for the purpose of studying eye diseases. But the Findings to Date section in the abstract took an odd and unexpected turn to talk about dementia. The introduction also starts out with dementia (imagine a reader reading the main text immediately after the title, it can get confusing) and only in the second paragraph it is clear that the eye imaging is mainly used for the dementia research. However, OCT is not used in dementia diagnosis, as the authors mentioned in the second paragraph. I suggest modifying the title to reflect the focus on dementia and/or make some changes to the introduction to manage readers' expectations.		Original: To describe the rationale and methodology of eye and vision assessments in the UK Biobank Repeat Imaging study. Revised: The retina provides biomarkers of neuronal and vascular health that offer promising insights into cognitive ageing, mild cognitive impairment (MCI) and dementia. This article described the rationale and methodology of eye and vision assessments with the aim of supporting the study of dementia in the UK Biobank Repeat Imaging study.
While it is implied in the title, perhaps indicate in the abstract whether the subset of participants returning had a baseline imaging. Also, which years the baseline imaging was conducted and whether the age (40–69) refers to baseline age.	We thank the reviewer for bringing out these two critical points. (1) We regret that the abstract cannot accommodate the details indicated in the manuscript's main text and tables due to word limits. UK Biobank participants who have completed their first imaging visit will be invited to attend a repeat set of scans (up to N=60,000). To clarify, all participants who returned to the repeat imaging study	(Line 44) Abstract Original: aged 40-69 enrolled across the UK Revised: aged 40-69 enrolled in 2006-2010 across the UK

	had baseline whole-body multimodal imaging data for the brain, heart, bones, and abdomen, but not all had baseline eye and vision-related data. Based on data from N=48,998 participants who had attended the baseline imaging assessment, approximately 28% had undergone retinal imaging (details shown in Table 2). (2) yes – ‘age (40-69)’ refers to baseline age and the baseline visit was conducted in 2006-2010. We’ll revise the abstract accordingly.	
Line 54 abstract: “... in 2022–2028.”	We thank the reviewer for pointing this out.	(Line 61) Abstract Original: will take place between 2022-2028. Revised: will take place in 2022-2028.
Last sentence of abstract and strengths of study: If spaces allows, I think the authors can afford to be a bit more optimistic. The repeat imaging studies are likely to benefit research in ophthalmic diseases and other health outcomes.	Thank you for this constructive suggestion. We will revise the abstract accordingly.	(Lines 61-63) The following sentence has been added to the abstract  ➤ Additionally, the broad and diverse data available in this study will support research into ophthalmic diseases and various other health outcomes beyond dementia.
Introduction “The eye provides insights into the risk or presence of all systemic diseases...” – I’m not sure this is accurate. It may be true for most major diseases but it’s hard to prove for “all” diseases (e.g., cancers?).	Thank you for your comment. We acknowledge that this statement - "The eye provides insights into the risk or presence of all systemic diseases" - might be exaggerated and require more clarification. While it is true that ocular manifestations indicate some major systemic diseases, such as diabetes and hypertension, we understand this may not apply to all conditions (e.g., cancers). We have revised this sentence to a more precise and accurate	(Lines 94-95) Original: The eye provides insights into the risk or presence of all systemic diseases, including hypertension and diabetes, Revised: The eye provides insights into the risk or presence of some major systemic diseases, including hypertension and diabetes,

	version. Once again, thank you for bringing this to our attention.	
“Using quantitative measurements from OCT scans, it may be possible to assess causal relationships between risk factors and retinal biomarkers related to dementia.” – It is not clear sure how OCT scans of the eye, even if of the retinal ganglion cells/nerve fibres or the retinal vasculature, can help assess these relationships. I’m assuming that the OCT measures will be used as surrogates of dementia? However, this will have the problem of factors that locally affect the ocular structures (e.g. IOP, refractive error) which are unlikely to impact the systemic outcome.	Thank you for your thoughtful comment. We agree that it may not be immediately clear how OCT scans of the eye can help assess the causal relationships between risk factors and retinal biomarkers related to dementia. We acknowledge the potential impact of factors that locally affect the ocular structures (e.g., intraocular pressure and refractive errors). In addition, whether OCT biomarkers could be a surrogate for dementia remains the subject of debate. We’ll revise the manuscript accordingly.	(Lines 122-123) Original: Using quantitative measurements from OCT scans, it may be possible to assess causal relationships between risk factors and retinal biomarkers related to dementia. Revised: Using quantitative measurements from OCT scans, it may be possible to assess causal relationships between risk factors and retinal biomarkers related to dementia.
It may be worth mentioning how the subset of the participants who underwent eye testing at baseline were selected (line 150). It is also unclear whether the 60,000 return participants have had a baseline OCT. It would be interesting to see how longitudinal change in retinal biomarkers is associated with cognitive measures and incidence of dementia.	Thank you for your valuable comment. (1) It would be helpful to provide more information on how the subset of participants who underwent eye testing at baseline was selected, and we will add this information to the Cohort Description section. (2) We’ll also clarify that only a subset of the 60,000 return participants had a baseline OCT examination. Based on the current data, there are roughly 28% of the participants in the Imaging Study had undergone retinal imaging (details shown in Table 2). The exact numbers of the 60,000 returning participants who had prior retinal imaging data is unknown, as there is	(1) Cohort Description section > UK Biobank subsection (Lines 152-158) Original: At the baseline assessment in 2006-2010, various eye measures including visual acuity, autorefraction, keratometry, intraocular pressure, corneal biomechanics, and retinal imaging comprising disc/macular digital colour photographs and a 3D macular OCT were performed on a subset of the UK Biobank participants – e.g., over 110,000 participants have completed the visual acuity, refractive error, and intraocular pressure measurements; and ~67,000 participants underwent retinal imaging. Revised: At the baseline visit, ophthalmic assessments were performed on a subset of participants between 2009-2010 at 6 of 22 UK Biobank assessment centres, including visual acuity, autorefraction,

	a policy for the UK Biobank Repeat Imaging Study to over-sample participants with baseline eye data. So, the estimated proportion in the repeat imaging study should be more than 28%.	keratometry, intraocular pressure, corneal biomechanics, and retinal imaging comprising disc/macular digital colour photographs and a 3D macular OCT. Over 110,000 participants completed the visual acuity, refractive error, and intraocular pressure measurements, and ~67,000 underwent retinal imaging. (2) See lines 410-411 in the revised manuscript - 'A subset of 13,732 (28%) participants had undergone retinal imaging.' The following sentence was added in lines 411-414 – As there is a policy for the UK Biobank Repeat Imaging Study to over-sample participants with baseline retinal imaging, the estimated numbers of participants with overlapping retinal imaging and whole-body imaging data in the repeat imaging visit will be more than 16,800.
Reviewer #2 Prof. Helmut Kuechenhoff, Ludwig-Maximilians-Universitat Munchen		
The UK Biobank is an important and well-designed study and extremely useful for many types of research. However, the part of statistical analysis should contain more concrete strategies of modelling those data. There are basically no strategies for longitudinal analysis of the data (p. 17) At least some basic ideas of longitudinal data analysis could be given in some detail.	We agree with the reviewer that the original manuscript lacked detailed statistical analysis plans, but we were trying to follow the authors' instructions not to report such information in a 'cohort profile' manuscript. Nonetheless, we appreciate this comment, as it has highlighted the significance of providing concrete strategies for modelling longitudinal data in this context. We have added some statistical content in the revised manuscript to address this important issue.	Cohort description section > Statistical Analysis Plan subsection: the following sentences were added in lines 378-386 of the revised manuscript:  ➤ The longitudinal nature of the data will allow models to be developed for incident cognitive outcomes / neurodegenerative events using multivariable Cox proportional hazards models with relevant eye measures (i.e., OCT, retinal vasculometry derived measures) as continuous predictors both with and without inclusion of other parameters, including age at cognitive decline /

		neurodegenerative onset, sex, ethnicity (although the cohort is largely of white European ancestry), smoking status (current, former and never), alcohol consumption, body mass index, blood pressure, blood biochemistry measures, social deprivation (by postcode), physical activity / sedentary behaviors, and relevant family history where available.
Furthermore, power calculations on page 16 were made without mentioning the relating statistical test (t-test ?).	We thank the reviewer for pointing this out. We have provided more details about the statistical test that had been used for power calculation.	(Lines 361-362) The following segment was added: Original: This large sample size, will have 99% power (alpha = 0.001) to detect at least 0.03 standard deviation change in the cognitive score[41] or brain measures[42] per 1 standard deviation increase in any retinal biomarker (RNFL or retinal vasculometric measure). Revised: This large sample size, will have 99% power (alpha = 0.001) to detect at least 0.03 standard deviation change in the cognitive score[41] or brain measures[42] (based on F-tests of linear regression coefficients from cross-sectional analyses) per 1 standard deviation increase in any retinal biomarker (RNFL or retinal vasculometric measure).

VERSION 2 – REVIEW

REVIEWER	Samantha Lee Lions Eye Institute, Centre for Ophthalmology and Visual Science, University of Western Australia
REVIEW RETURNED	19-May-2023
GENERAL COMMENTS	The authors have addressed all of my comments. I look forward to research published from this imaging follow-up.